# Molecular Mechanisms of Takotsubo Syndrome

**DOI:** 10.3390/ijms232012262

**Published:** 2022-10-14

**Authors:** Liam S. Couch, Keith Channon, Thomas Thum

**Affiliations:** 1Department of Cardiovascular Medicine, University of Oxford, Oxford OX1 2JD, UK; 2Institute of Molecular and Translational Therapeutic Strategies, Hannover Medical School, 30625 Hannover, Germany; 3Fraunhofer Institute of Toxicology and Experimental Medicine, 30625 Hannover, Germany

**Keywords:** Takotsubo syndrome, adrenaline, β_2_AR, inflammation, metabolism, microvascular dysfunction

## Abstract

Takotsubo syndrome (TTS) is a severe but reversible acute heart failure syndrome that occurs following high catecholaminergic stress. TTS patients are similar to those with acute coronary syndrome, with chest pain, dyspnoea and ST segment changes on electrocardiogram, but are characterised by apical akinesia of the left ventricle, with basal hyperkinesia in the absence of culprit coronary artery stenosis. The pathophysiology of TTS is not completely understood and there is a paucity of evidence to guide treatment. The mechanisms of TTS are thought to involve catecholaminergic myocardial stunning, microvascular dysfunction, increased inflammation and changes in cardiomyocyte metabolism. Here, we summarise the available literature to focus on the molecular basis for the pathophysiology of TTS to advance the understanding of the condition.

## 1. Introduction

Takotsubo syndrome (TTS) is a severe but reversible acute heart failure syndrome that results from catecholaminergic stress [1,2,3]. TTS patients typically present with chest pain and dyspnoea, and often have ST-segment elevation on electrocardiogram (ECG) [4]. Consequently, TTS patients are often initially suspected to have acute myocardial infarction (AMI), and 1–2% of patients with suspected acute coronary syndrome (ACS) [5,6] and up to 5–6% of female patients presenting with suspected STEMI [7] are eventually diagnosed with TTS.

TTS characteristically involves apical akinesia of the left ventricle, with concomitant basal hyperkinesia. Whilst this may involve other patterns of contractile dysfunction in a minority of cases, the acute contractile dysfunction in TTS occurs in the absence of culprit coronary artery disease [4]. TTS predominantly affects post-menopausal women, and typically occurs following extreme physical or emotional stress [2,4]. Serious complications result from the extreme ventricular dilatation and reduced cardiac output, including cardiogenic shock, thrombi formation, pulmonary oedema, arrhythmia generation and LV rupture. TTS is associated with significant mortality burden, with short- and long-term mortality similar to AMI [7,8]. Patients also experience long-term contractile dysfunction [9], and are at approximately a 10% risk of recurrence [10].

Since there is no evidence-based treatment for TTS, understanding its pathophysiology is important. Here, we summarise the available literature by performing searches into TTS and the adrenergic system, microvascular dysfunction, oestrogen, genetic changes and biomarkers to produce this review focussing on the molecular mechanisms of TTS.

## 2. Cardiomyocyte Contractility and the Adrenergic System

Catecholamines are evidenced to play a causative role in TTS. TTS has been frequently reported to occur secondary to medical conditions with elated catecholamine levels such as pheochromocytoma [11,12,13], acute subarachnoid haemorrhage [14,15] and acute thyrotoxicosis [16]. Further, iatrogenic TTS occurs in stress dobutamine echocardiography and after adrenaline administration [17,18]. Adrenaline levels were reported by Wittstein et al. to be 10 to 20 times normal in 13 patients with TTS, and also higher than in STEMI [19]. Pre-clinical models of TTS provide additional evidence, as TTS can be robustly induced in rodent and primate models with adrenaline [20,21]. More recent studies, however, have failed to demonstrate elevated catecholamine levels in TTS patients [22], although circulating catecholamine levels are extremely short lived [23].

The central involvement of β-adrenergic receptors (βARs) was suggested by Lyon et al. to underpin the pattern of regional wall motion abnormalities (RWMA) that occur in TTS [24]. In normal physiological conditions, βARs modulate rate and force of myocardial contraction and relaxation, allowing response to stress or exercise. This process becomes dysregulated in disease, and beta-blockade is a mainstay of heart failure therapy.

βARs are spatially organized across the left ventricle (LV), with the highest density in the LV apex [20,25,26] which become exposed to circulating adrenaline. In contrast, the base of the LV has a greater density of sympathetic nerve terminals [27] and tends to receive stimulation via noradrenaline. In particular, the human myocardium demonstrates a higher concentration of β_2_AR when compared to other mammals (β_1_AR to β_2_AR ratio of 4:1), with minimal expression of β_3_AR [28,29]. Indeed, this apicobasal difference is demonstrated by larger apical βAR response in vivo [30,31,32] and β_2_AR response in vitro [20,33], which has been demonstrated to be consequent to fewer caveolae (sequestration) and lower cAMP buffering by PDEs [33].

Both β_1_AR and β_2_AR signal via the canonical stimulatory Gαs pathway, which leads to activation of the enzyme adenylyl cyclase (AC) and increases intracellular cAMP [34,35]. The intracellular cAMP compartmentalisation of these respective adrenoceptors differ, with β_1_AR-Gαs resulting in cell wide increases in cAMP whereas β_2_AR-Gαs results in localised signalling. cAMP subsequently activates PKA which phosphorylates downstream protein targets to result in positive inotropy, lusitropy and chronotropy.

At higher agonist concentrations, the pleiotropic β_2_AR can also signal via the inhibitory Gαi [36,37], and physiologically, this limits the pro-apoptotic and pro-arrhythmic toxic effects of Gαs activity [38]. This shifts receptor coupling to Gαi, in a process known as stimulus trafficking or biased agonism [36]. Signalling via cAMP-PKA phosphorylates the β_2_AR and primes the receptor for subsequent phosphorylation by G-protein coupled receptor kinase (GRK) whereby GRK is recruited by Gβγ [39]. This results in internalisation of the β_2_AR and stimulus trafficking to β_2_AR-Gαi in a two-step process. This directly and indirectly opposes Gαs activity and is negatively inotropic by downregulating cAMP activity and activating further signalling pathways including p38-MAPK [40].

Excess circulating adrenaline release in TTS may result in extreme negative inotropy via the β_2_AR-Gαi [24], which typically results in apical hypokinesia owing to the gradient of adrenergic receptors within the heart as previously discussed. This hypothesis is supported by an in silico modelling study that recreated Takotsubo-like contractile dysfunction with apical dysfunction following intense agonist stimulation when apical-basal β_1_AR and β_2_AR gradients were introduced [41]. Further, TTS-like dysfunction can be induced in vivo by bolus injection adrenaline or isoprenaline [20,21,42,43,44]. Interestingly, the hyperthermic response subsequent to catecholamine administration seems necessary for the induction of TTS, since a recent in vivo study demonstrated that maintenance of hyperthermia would induce TTS-like contractile dysfunction following isoprenaline administration, whereas active cooling prevented, but did not reverse, these changes [43].

The pivotal role of β_2_AR-Gαi in TTS induction has been demonstrated in vivo, with adrenaline-induced negative inotropy prevented with Gαi inhibition using pertussis toxin (PTX) in vivo and in vitro [20]. However, as in resting physiology, β_2_AR-Gαi appears cardioprotective in TTS, as complete inhibition with β_2_AR antagonists or PTX prevents apical hypokinesis but also increases mortality by approximately 70% [42]. Further, specific inhibition of the β_2_AR-Gαi-p38MAP kinase pathway in vivo TTS also increases mortality [20].

Changes in G-protein expression were noted with upregulation of TTS-associated microRNAs (miR-16 and miR-26a), whereby predisposition to TTS generation was noted in vivo following epinephrine administration [44]. This was associated with reductions in RGS4 (regulator of G-protein signalling 4) and G-protein subunit Gβ (GNB1); [44] proteins which are involved in the termination of β_2_AR-Gαi signalling in resting physiology. This potentiated Gαi activity was noted in the reduction in baseline contractility of apical but not basal rat cardiomyocytes with upregulation of TTS-associated microRNAs, and was reversed on inhibition of Gαi with PTX [44].

Nef et al. analysed endomyocardial biopsy samples from 16 patients with TTS to show increased activation of PI3K and Akt that was found to be normal at follow-up and is absent in ischemia [45]. These pathways are linked to β_2_AR-Gαi activity and promote cardiomyocyte survival. More recently, Nakano et al. observed proteins involved in β_2_AR-Gαi signalling to be increased in tissue samples from 26 patients with acute TTS, including increased levels of GRK and β-arrestin [46], proteins that are necessary for β_2_AR-Gαi stimulus trafficking to occur.

Changes in contractility may also occur via direct changes in cardiomyocyte calcium handing. Reduced calcium transient amplitude, SR calcium content and peak calcium current amplitude were found in apical cardiomyocytes when TTS-associated miRs were upregulated, and this was found to consequent from downregulation of CACNB1 (L-type calcium channel Cavβ subunit) [44], a channel that controls LTCC current [47]. Conversely, increased calcium transient amplitude and kinetics were observed in an induced-pluripotent stem cell derived cardiomyocytes (iPSC-CMs) line produced from TTS patients [48]. This was observed alongside increased β_1_AR, β_2_AR and cAMP responses, altered metabolism and increased intracellular lipid accumulation [48]. This TTS cell line demonstrated reduced baseline force generation when engineered heart tissue (EHT) strips of iPSC-CMs were produced, and these had increased sensitivity and reduced desensitization to catecholamines [48].

## 3. Vascular Dysfunction

After its identification, TTS was initially suggested to result from multivessel coronary vasospasm [49,50]. Subsequently, it has been linked to vasomotor dysfunction with increased vascular reactivity and altered endothelial function following psychological stress in patients with previous TTS [51]. Microvascular dysfunction has also been demonstrated in cases of TTS [52,53] which has the potential to better explain the phenotype of acute ischaemic stunning [54]. Early studies found myocardial contrast echocardiography demonstrated myocardial perfusion abnormality in patients with acute TTS which was partially reversible by adenosine and not present after 1 month, unlike in STEMI where the perfusion defect is fixed [55]. However, induction of TTS-like cardiac dysfunction in rat with isoprenaline produces myocardial dysfunction without preceding myocardial perfusion defects [56]. As such, vascular or microvascular dysfunction observed may be a consequence of the catecholamine surge in TTS. Interestingly, the CIRCUS-TTS study did not find any differences in systemic microvascular function between TTS, MI or healthy control both acutely or after 3 month follow-up [57].

The catecholamine surge that occurs in TTS likely leads to endothelial dysfunction [58] which could sensitise to vasospasm upon provocation. Prevalence of vascular dysfunction seems variable in TTS patients and has often been found to be absent [59,60,61]. Endothelin is increased in TTS and could mediate hypothetical vasospasm, however endothelin is increased to the same degree in STEMI [62] where the phenomenon of catecholamine-induced contractile dysfunction does not occur. Indeed, catecholamines such as adrenaline or dobutamine [17] vasodilate coronary arteries, and preclinical models inducing TTS with catecholamine in the absence endothelin cause apical dysfunction [20,42] in the absence of myocardial perfusion abnormality [56]. Given a diagnostic criterion of TTS is the presence of contractile abnormality usually occurring over more than one coronary territory [63], phenotypes of TTS would not be explained by single or multivessel spasm, and endomyocardial biopsies show absence of ischaemic, stunned or hibernating myocardium [54,64]. Since TTS is similar in presentation to ACS, it has been suggested that TTS is a form of microvascular or aborted MI [65]. Despite this, intravascular ultrasound studies in acute TTS have not found plaque rupture, endothelial breach or intracoronary thrombus [66,67], and there is no difference in incidence of wrap-around left anterior descending coronary artery in TTS [68]. Further, long-term recurrence of TTS is not influenced by aspirin therapy [69].

## 4. Oestrogen Deprivation

Oestrogen confers cardioprotection via a multitude of mechanisms, including sympatholysis [70,71], and 80% of TTS patients are post-menopausal women [4]. Basal circulating adrenaline is lower in women than in men [72], as is urinary cortisol, adrenaline and noradrenaline, which rise with age [73]. Interestingly, hormone replacement therapy reduced the levels of these stress hormones, suggesting regulation by oestrogen [73]. However, there is no difference in oestrogen levels between female TTS patients and age-matched patients with MI [74]. Women demonstrate greater vascular β_2_AR sensitivity [75] which suggests oestrogen may be able to potentiate β_2_AR signalling. Oestrogen can signal by downstream β_2_AR-Gαi-PI3K and -Akt pro-survival pathways [76] which confer cardioprotective effects [24] as earlier discussed.

However, agonist stimulation of the G protein-coupled oestrogen receptor (GPER) with G1/E2 prevented the TTS-like contractile changes elicited by high dose adrenaline in rodent [77]. In vitro, G1/E2 reduced phosphorylation and internalization of β_2_AR [77], and Gαi activity, and increased cAMP concentration in cardiomyocytes treated with adrenaline [77]. Further, in these experiments G1/E2 prevented the decreased Ca2+ amplitude and channel current (ICa-L) caused by adrenaline in rat cardiomyocytes [77].

Adrenergic dependent gene expression changes in immediate early genes (IEGs) in endothelial cells, myocardial cells and coronary smooth muscle cells, including c-fos but not c-jun, rapid activation of p44/p42 MAP kinase and heat shock protein 70, are observed in a rodent model of emotional stress causing TTS by conscious immobilization [78,79], and are prevented by oestrogen supplementation [80]. A potent vasodilator failed to inhibit expression of IEG mRNAs suggesting these changes are not driven by ischemia [78]. Oophorectomy promotes sympathetic activity by upregulating β_1_AR expression, which is reversed by subsequent reintroduction of oestrogen [81]. Further, β_1_AR upregulation caused by exposure to catecholamines and following ischemia-reperfusion injury is reduced by chronic oestrogen exposure [82]. The impact of hormone replacement therapy in post-menopausal women on TTS incidence and recurrence is not clear, and offers an interesting avenue for further investigation and possible prevention of TTS.

## 5. Genomic Changes

Efforts have been made to identify genetic mutations in patients with TTS, however these are poorly conserved and relatively few have been found. The L41Q GRK5 ‘gain-of-function’ polymorphism in transgenic mice and isolated cells results in enhanced cardiac GRK5 activity, βAR phosphorylation and βAR desensitization, and causes negative inotropy after high catecholamine release [83]. Whilst GRK5 polymorphism was found to be more common in two Italian TTS cohorts [84,85], this finding has not been demonstrated elsewhere [86]. Although examples of family predilection to TTS have been noted in several case series [87,88,89,90], specific genetic mutations have not been identified. Several functional polymorphisms have been proposed and identified in isolated cases for α_1_AR, β_1_AR, β_2_AR, GRK5 and oestrogen receptors [91]. Further, whole exome sequencing of the aforementioned TTS iPSC-CM patient line did not identify any genetic differences [48].

An additional source of genomic variation may exist in different disease states from non-coding RNA (ncRNA), which can be utilised both in biomarker detection [92] and in understanding disease pathophysiology. miR-1, miR-16, miR-26a and miR-133a are significantly raised in TTS patients when compared to healthy control [62]. However, the profile of these differences enabled differentiation from ST-elevation myocardial infarction (STEMI), which TTS is often confused for at patient presentation. miR-1 and miR-133a are significantly higher in STEMI whereas miR-16 and miR-26a were only significantly raised in TTS, and not STEMI [62]. These ‘TTS-associated miRs’ (miR-16 and miR-26a) have recently been identified to be involved in the pathophysiology of TTS, where they sensitise to TTS-like changes in contractility in vitro and in vivo following adrenaline, and cause TTS to be induced in vivo at lower adrenaline concentrations [44]. In full, miR-16 and/or miR-26a upregulation in isolated apical, but not basal, adult rat cardiomyocytes reduce baseline contractility [44]. This was reversed with PTX (as discussed above) and reproduced in non-failing human cardiomyocytes [44]. Sensitivity to adrenaline was reduced in apical cardiomyocytes with increased TTS-miRs, and the positive inotropic effect of adrenaline was increased only in basal cardiomyocytes [44]. These changes occurred via reduction in protein level of CACNB1 (L-type calcium channel Ca_v_β subunit), RGS4 (regulator of G-protein signalling 4) and G-protein subunit Gβ (GNB1) [44] as discussed above.

Mutation in Bcl2-associated athanogene 3 (BAG3) 3′-UTR (a component of the chaperone-assisted autophagy pathway) has been noted in a TTS patient cohort that prevented miR-371a-5p binding and increased level of BAG3 in cardiomyocytes following exposure to adrenaline [93]. However, this report suggests increased protein expression of BAG3 following miR-371a binding which represents an atypical mechanism of miR regulation, and it remains unclear how changes in BAG3 contribute in TTS.

## 6. Inflammatory Signalling

Beyond the initial contractile dysfunction present in TTS, long-term sequelae following high adrenaline appear to involve cardiac inflammation. Endomyocardial biopsy from TTS patients demonstrate mononuclear infiltrates and contraction-band necrosis [19], and slowly resolving myocardial oedema is present on cardiac magnetic resonance imaging (CMR) [94]. As the acute oedema subsides, global microscopic fibrosis develops and is detected from 4 months [9]. This occurs alongside gene changes in metabolic pathways with a shift to lipid based metabolism [95]. This is associated with symptomatic and functional impairment after greater than 1 year following diagnosis of TTS, and is associated with persistent subclinical cardiac dysfunction [96].

Presence of nitrosative stress has been identified by immunohistological studies in LV myocardium from TTS [97], and altered nitric oxide (NO) signalling has also been demonstrated [98]. In vivo induction of TTS with isoprenaline in rodent has demonstrated increased CD68+ macrophages and levels of inflammatory markers [99]. Whilst inhibition of PARP-1 (Poly [ADP-ribose] polymerase 1) partially reversed apical radial strain and fractional shortening [99], this has no effect on inflammatory markers or oxidative stress [100].

TTS patients have also been observed to have a greater retention of ultra-small iron oxide superparamagnetic particle (USPIO - phagocytosed by activated tissue macrophages) which was not detectable at 5 month follow-up, and therefore their increased activity drives the increased myocardial inflammation in TTS [101]. Interleukin-6 and chemokine (C-X-C motif) ligand 1 were increased in this context, and macrophage subtype was shifted from CD14++CD16+ and non-classic monocytes to CD14++CD16- [101]. TTS induced in rodent with isoprenaline demonstrates localised myocardial inflammatory changes followed by clusters of predominantly M1 proinflammatory macrophages, a finding seen in post-mortem myocardial samples from TTS patients [102]. Interestingly this study found M2 macrophage levels to be correlated with positive recovery of cardiac function [102]. Whilst this inflammatory change has been observed in vivo following direct administration of catecholamines [102], TTS has been triggered by immune checkpoint inhibitors which activate T lymphocytes and increase inflammation [103,104].

## 7. Mitochondrial Dysfunction

Alongside apical contractile dysfunction seen in rats treated with high dose catecholamines, intracellular lipid deposition is seen which supports the presence of acute mitochondrial dysfunction [42], as well as apoptosis [105] and fibrosis [106]. Similar findings have been observed in clinical TTS [64,107]. Comparison of TTS iPSC-CMs also observed increased intracellular lipid accumulation and reduced mitochondrial function [48].

Recent data from an isoprenaline-induced TTS model shows significant alterations in glucose and fatty acid metabolism as well as Krebs cycle activity in TTS [108]. Indeed, cardiac 18F-FDG metabolic rate was found to be increased in TTS in rodent, and expression of GLUT4-RNA/GLUT1/HK2-RNA was increased, with accumulation of glucose- and fructose-6-phosphates and increased hexokinase activity [108]. There was a shift from lactate and pyruvate to β-Oxidation enzymes CPT1b-RNA and 3-ketoacyl-CoA thiolase, and although malonyl-CoA (CPT-1 regulator) activity was seen to be increased, fatty acids and acyl-carnitines levels were reduced [108]. Krebs cycle intermediates α-ketoglutarate and succinyl-carnitine, along with dihydroorotate (a cellular ATP reporter) were reduced [108]. Mitochondrial Ca2+ uptake was initially impaired, inducing oxidation of NAD(P)H and FAD [108].

## 8. Brain Heart Axis

TTS was shown to be closely associated with neuropsychiatric disorders by Templin et al. from their study of 1750 patients with TTS [4], and there seems to be a clear interaction between brain and heart in this context. Whilst original data looked at various neurological and psychiatric conditions [4], more recent literature review suggested that pre-admission anxiety disorders were the most prevalent abnormalities in patients presenting with acute TTS [109]. TTS-associated miRs recently [62] have also been shown to be altered in neuropsychiatric stress [110,111,112]. Consequently it is interesting that they are involved in the predisposition to TTS generation in rodent as above [44], and this suggests that chronic neuropsychiatric stress may predispose to future generation of TTS.

MRI studies have assessed the role of brain regions associated with the autonomic nervous system (ANS) and emotion, with altered brain architecture being observed in patients following TTS. Specifically, reduced insula and cingulate cortex thickness has been noted, alongside reduced connectivity in the limbic system and ANS-specific network [113]. This included alterations in the left amygdala, hippocampi, left para-hippocampal gyrus, left superior temporal pole, and right putamen [113]. Both parasympathetic and sympathetic systems are altered, suggesting the balance within the ANS may be more important than the sympathetic activity alone [114]. Indeed, there is hypoconnectivity of central brain regions associated with autonomic functions and regulation of the limbic system in patients with TTS [115]. Whether this is a cause or consequence of the large catecholamine burden experienced by TTS patients is unknown. This could explain the ‘gain’ in the hypothalamic-pituitary axis as previously suggested [2,19].

An interesting study that retrospectively identified individuals who underwent clinical 18 F-FDG-PET/CT imaging prior to generation of TTS found that TTS patient had higher baseline amygdalar activity, and amygdalar activity was associated with the risk of subsequent TTS generation [116]. Further, TTS patients with higher amygdalar activity were likely to develop TTS approximately 2 years sooner that those with low activity [116].

Neurogenic stunned myocardium (NSM) represents a condition that demonstrates characteristic myocardial contractile dysfunction following increased sympathetic activity after neurological insult. It has therefore been compared to TTS [117,118,119,120]. Whilst NSM is observed to have higher catecholamines levels, echocardiographic findings are similar to TTS [119]. Although there are minor differences in presentation [121], NSM could be a specific subset of TTS consequent from increased neurological sympathetic activity as opposed to the classical increase in circulating catecholamine that occurs in TTS. The conditions have previously been compared in great detail to conflicting opinions [119,120,122].

## 9. Biomarkers

Owing to the modest ischaemic damage and profound contractile abnormality present in TTS, modest increases in creatine kinase-MB and cardiac troponin are seen when compared to AMI, versus the significant elevation of natriuretic peptides, including brain natriuretic peptide (BNP) or N-terminal-pro-BNP (NT-pro-BNP) [123,124,125]. Indeed, admission NT-pro-BNP levels have been shown to act as an independent predictor of morbidity and mortality [126].

TTS-associated miRs have been discussed above, whereby miR-1, miR-16, miR-26a and miR-133a are increased in TTS compared to healthy patients, but miR-1 and miR-133a are higher in STEMI than TTS, whereas miR-16 and miR-26a were raised only in TTS [62]. The potential importance that biomarkers play in understanding the pathophysiology of TTS was illustrated in a follow-up study, showing miR-16 and miR-26a were involved in the predisposition to TTS generation in rodent as above [44].

Given the previously discussed involvement of inflammation and microvascular dysfunction in TTS [1,3], markers for these have been investigated for use as biomarkers. IL-2, IL-4, IL-10, IFN-γ and TNF-α have been seen to be higher in acute TTS than in AMI, with IL-6 being higher in AMI [127]. Further, co-peptin and endothelin, two vasocontricting peptides, are altered in TTS when compared to AMI. Co-peptin levels are substantially increased in AMI but normal or only modestly elevated in TTS [128,129,130]. When compared to control patients, endothelin levels are increased to a similar level in TTS and STEMI [62], and glycocalyx levels also seem elevated in TTS [131].

## 10. Conclusions

The molecular mechanisms of TTS are becoming further elucidated. There is a clear pathogenic link to a catecholamine surge which causes changes in cardiomyocyte contractility through the beta-adrenergic system. Oestrogen withdrawal post-menopause plays a role in predisposing patients to developing TTS, and it seems that chronic stress may also increase the future likelihood of TTS following acute stress. Circulating adrenaline also acts on the non-myocyte cellular population within the heart which likely results in microvascular dysfunction, however it is unclear if this contributes to contractile dysfunction. Further mechanisms include changes in cardiomyocyte metabolism and persistent inflammation which likely contributes to long-term cardiac dysfunction.

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
