# Peer review of "Molecular Mechanisms of Takotsubo Syndrome"

_ijms, 2022, doi:10.3390/ijms232012262_

Round 1

Reviewer 1 Report

I agree with the authors that Takotsubo syndrome (TTS) is still a mysterious disease. TTS carries a heavy burden of mortality, with both short-term and long-term mortality, and we do not have a causal treatment for this disease.

1.Title and abstract

The title and abstract is consistent with the presented problem and reflects the main message of the review.

2. Introduction

Introduction is clear and helpful to readers unfamiliar with the problem.

3. Review
The authors presented the current state of knowledge regarding the pathomechanism of TTS very well. The mechanisms related to the adrenergic system have been described particularly precisely.

4. Conclusions
The conclusions are logical.

Irregularities:

1.      Each first abbreviation should be expanded. For example, the abbreviation ECG has no expansion.

2.      Incorrect vocabulary: “ST segment” should be used instead of “ST”.

The paper is interesting and worth publishing after small corrections.

Author Response

Many thanks for your kind comments. The irregularities you highlight have been corrected.

Reviewer 2 Report

I would like to congratulate the authors for their interesting and informative paper.

This is a narrative review on the molecular mechanisms that underlie Takotsubo syndrome. Specifically, the authors discuss catecholaminergic myocardial stunning, microvascular dysfunction, increased inflammation, and changes in cardiomyocyte metabolism as potential contributors to the pathogenesis of this syndrome.

This is an overall well written manuscript. Here, I have made a few suggestions that (in my opinion) could help to further improve the quality of the paper.

·         The authors may consider describing the literature search in order to be transparent about the sources of information on which their article is based. Although it is not necessary to describe the literature search in as much detail as for a systematic review, specifying search terms and inclusion criteria (types of literature included) may be important.

·         The authors may consider providing more information about the referenced studies when presenting evidence of key arguments (for instance, about the histopathologic alterations observed in Takotsubo cardiomyopathy).

·         The authors may consider presenting effect sizes when reporting data from relevant studies.

Author Response

Many thanks for your comments and valuable suggestions. Whilst this is not a systematic review and therefore does not have a methods section, we have elaborated on the search strategy within the introduction. We have also added author names and patient numbers in instances where key information is discussed, such as in the human biopsy section (as suggested) and elsewhere.

Reviewer 3 Report

Dear Authors, 

I read your work with interest and I think that it is excellent. Probably you could add some pictures and tables to improve the understanding. 

Best Regards

Author Response

Many thanks for your kind comments. Unfortunately, in the short time provided for revision we were unable to create a schematic of sufficient quality to include, and felt the article processing charges that would be associated with including figures from other publications would not be possible for this open access article. We hope you feel this review of sufficient standard for publication in IJMS without further figures.

Round 2

Reviewer 2 Report

Thank you very much for considering my comments and suggestions and revising your manuscript accordingly.